# SYNTHNET: LEARNING SYNTHESIZERS END-TO-END

## ABSTRACT

Learning synthesizers and generating music in the raw audio domain is a challenging task. We investigate the learned representations of convolutional autoregressive generative models. Consequently, we show that mappings between musical notes and the harmonic style (instrument timbre) can be learned based on the raw audio music recording and the musical score (in binary piano roll format). Our proposed architecture, SynthNet uses minimal training data (9 minutes), is substantially better in quality and converges 6 times faster than the baselines. The quality of the generated waveforms (generation accuracy) is sufficiently high that they are almost identical to the ground truth. Therefore, we are able to directly measure generation error during training, based on the RMSE of the Constant-Q transform. Mean opinion scores are also provided. We validate our work using 7 distinct harmonic styles and also provide visualizations and links to all generated audio.

## 1 INTRODUCTION

WaveNets (Van Den Oord et al., 2016) have revolutionized text to speech by producing realistic human voices. Even though the generated speech sounds natural, upon a closer inspection the waveforms are different to genuine recordings. As a natural progression, we propose a WaveNet derrivative called SynthNet which can learn and render (in a controlled way) the complex harmonics in the audio training data, to a high level of fidelity. While vision is well established, there is little understanding over what audio generative models are learning. Towards enabling similar progress, we give a few insights into the learned representations of WaveNets, upon which we build our model.

WaveNets were trained using raw audio waveforms aligned with linguistic features. We take a similar approach to learning music synthesizers and train our model based on the raw audio waveforms of entire songs and their symbolic representation of the melody. This is more challenging than speech due to the following differences: 1) in musical compositions multiple notes can be played at the same time, while words are spoken one at a time; 2) the timbre of a musical instrument is arguably more complex than speech; 3) semantically, utterances in music can span over a longer time.

Van Den Oord et al. (2016) showed that WaveNets can generate new piano compositions based on raw audio. Recently, this work was extended by Dieleman et al. (2018), delivering a higher consistency in compositional styling. Closer to our work, Engel et al. (2017) describe a method for learning synthesizers based on individually labelled note-waveforms. This is a labourius task and is impractical for creating synthesizers from real instruments. Our method bypassses this problem since it can directly use audio recodings of an artist playing a given song, on the target instrument.

SynthNet can learn representations of the timbre of a musical instrument more accurately and efficiently via the dilated blocks through depthwise separable convolutions. We show that it is enough to condition only the first input layer, where a joint embedding between notes and the corresponding fundamental frequencies is learned. We remove the skip connections and instead add an additional loss for the conditioning signal. We also use an embedding layer for the audio input and use SeLU (Klambauer et al., 2017) activations in the final block.

The benchmarks against the WaveNet (Van Den Oord et al., 2016) and DeepVoice (Arik et al., 2017) architectures show that our method trains faster and produces high quality audio. After training, SynthNet can generate new audio waveforms in the target harmonic style, based on a given song which was not seen at training time. While we focus on music, SynthNet can be applied to other domains as well.

**Our contributions are as follows: 1)** We show that musical instrument synthesizers can be learned end-to-end based on raw audio and a binary note representation, with minimal training data. Multiple instruments can be learned by a single model. **2)** We give insights into the representations learned by dilated causal convolutional blocks and consequently propose SynthNet, which provides substantial improvements in quality and training time compared to previous work. Indeed, we demonstrate (Figure 4) that the generated audio is practically identical to the ground truth. **3)** The benchmarks against existing architectures contains an extensive set of experiments spanning over three sets of hyperparameters, where we control for receptive field size. We show that the RMSE of the Constant-Q Transform (RMSE-CQT) is highly correlated with the mean opinion score (MOS). **4)** We find that reducing quantization error via dithering is a critical preprocessing step towards generating the correct melody and learning the correct pitch to fundamental frequency mapping.

## 2 RELATED WORK

In music, style can be defined as the holistic combination of the melodic, rhythmic and harmonic components of a particular piece. The delay and sustain variation between notes determines the *rhythmic style*. The latter can vary over genres (e.g. Jazz vs Classical) or composers. Timbre or *harmonic style* can be defined as the short term (attack) and steady state (sustained frequency distribution) acoustical properties of a musical instrument (Sethares, 2005). Our focus is on learning the *harmonic style*, while controlling the (given) melodic content and avoiding any rhythmic variations.

The research on content creation is plentiful. For an in depth survey of deep learning methods for music generation we point the reader to the work of Briot et al. (2017). Generative autoregressive models were used in (Van Den Oord et al., 2016; Mehri et al., 2016) to generate new random content with similar harmonics and stylistic variations in melody and rhythm. Recently, the work of Van Den Oord et al. (2016) was extended by Dieleman et al. (2018) where the quality is improved and the artificial piano compositions are more realistic. We have found piano to be one of the easier instruments to learn. Donahue et al. (2018) introduce WaveGANs for generating music with rhythmic and melodic variations.

Closer to our work, Engel et al. (2017) propose WaveNet Autoencoders for learning and merging the harmonic properties of instrument synthesizers. The major difference with our work is that we are able to learn harmonic styles from entire songs (a mapped sequence of notes to the corresponding waveform), while their method requires individually labelled notes (NSynth dataset). With our method the overhead of learning a new instrument is greatly reduced. Moreover, SynthNet requires minimal data and does not use note velocity information.

Based on the architecture proposed by Engel et al. (2017), and taking a domain adaptation approach, Mor et al. (2018) condition the generation process based on raw audio. An encoder is used to learn note mappings from a source audio timbre to a target audio timbre. The approach can be more error prone than ours, since it implies the intermediary step of correctly decoding the right notes from raw audio. This can significantly decrease the generation quality. Interestingly, Mor et al. (2018) play symphonic orchestras from a single instrument audio. However, there is no control over which instrument plays what. Conversely, we use individual scores for each instrument, which gives the user more control. This is how artists usually compose music.

## 3 END-TO-END SYNTHESIZER LEARNING

Van Den Oord et al. (2016) and Arik et al. (2017) have shown that generative convolutional networks are effective at learning human voice from raw audio. This has advanced the state of the art in text to speech (TTS). Here, we further explore the possibilities of these architectures by benchmarking them in the creative domain – learning music synthesizers. There are considerable differences between the human voice and musical instruments. Firstly, the harmonic complexity of musical instruments is higher than the human voice. Second, even for single instrument music, multiple notes can be played at the same time. This is not true for speech, where only one sound utterance is produced at a time. Lastly, the melodic and rhythmic components in a musical piece span a larger temporal context than a series of phonemes as part of speech. Therefore, the music domain is much more challenging.

### 3.1 BASELINE ARCHITECTURES

The starting point is the model proposed by Van Den Oord et al. (2016) with the subsequent refinements in (Arik et al., 2017). We refer the reader to the these articles for further details. Our data consists of triplets $\{(\boldsymbol{x}_1, \boldsymbol{y}_1, \boldsymbol{z}_1), \ldots, (\boldsymbol{x}_N, \boldsymbol{y}_N, \boldsymbol{z}_S)\}$ over $N$ songs and $S$ styles, where $\boldsymbol{x}_i$ is the 256-valued encoded waveform, $\boldsymbol{y}_i$ is the 128-valued binary encoded MIDI and $\boldsymbol{z}_s \in \{1, 2, \ldots, S\}$ is the one-hot encoded style label. Each audio sample $x_t$ is conditioned on the audio samples at all previous timesteps $\boldsymbol{x}_{<t} = \{x_{t-1}, x_{t-2}, \ldots, x_1\}$, all previous binary MIDI samples and the global conditioning vector. The joint probability of a waveform $\boldsymbol{x} = \{x_1, \ldots, x_T\}$ is factorized as follows:

$$p(\boldsymbol{x}|\boldsymbol{y}, \boldsymbol{z}) = \prod_{t=1}^{T} p(\boldsymbol{x}|\boldsymbol{x}_{<t}, \boldsymbol{y}_{<t}, \boldsymbol{z}). \tag{1}$$

The hidden state before the residual connection in dilation block $\ell$ is

$$\boldsymbol{h}^\ell = \tau\Big(\boldsymbol{W}_f^\ell * \boldsymbol{x}^{\ell-1} + \boldsymbol{V}_f^\ell * \boldsymbol{y}^{\ell-1} + \boldsymbol{U}_f^\ell \cdot \boldsymbol{z}\Big) \odot \sigma\Big(\boldsymbol{W}_g^\ell * \boldsymbol{x}^{\ell-1} + \boldsymbol{V}_g^\ell * \boldsymbol{y}^{\ell-1} + \boldsymbol{U}_g^\ell \cdot \boldsymbol{z}\Big), \tag{2}$$

while the output of every dilation block, after the residual connection is

$$\boldsymbol{x}^\ell = \boldsymbol{x}^{\ell-1} + \boldsymbol{W}_r^\ell \cdot \boldsymbol{h}^\ell, \tag{3}$$

where $\tau$ and $\sigma$ are respectively the tanh and sigmoid activation functions, $\ell$ is the layer index, $f$ indicates the filter weights, $g$ the gate weights, $r$ the residual weights and $\boldsymbol{W}$, $\boldsymbol{V}$ and $\boldsymbol{U}$ are the learned parameters for the main, local conditioning and global conditioning signals respectively. The $f$ and $g$ convolutions are computed in parallel as a single operation (Arik et al., 2017). All convolutions have a filter width of $F$. The convolutions with $\boldsymbol{W}^\ell$ and $\boldsymbol{V}^\ell$ are dilated.

To locally condition the audio signal, Van Den Oord et al. (2016) first upsample the $\boldsymbol{y}$ time series to the same resolution as the audio signal (obtaining $\boldsymbol{y}^\ell$) using a transposed convolutional network, while Arik et al. (2017) use a bidirectional RNN. In our case, the binary midi vector already has the same resolution.

We use an initial causal convolution layer (Equation 4) that only projects the dimensionality of the signal from 128 channels to the number of residual channels. The first input layers are causal convolutions with parameters $\boldsymbol{W}^0$ and $\boldsymbol{V}^0$ for the waveform and respectively the piano roll:

$$\boldsymbol{y}^\ell = \boldsymbol{V}^0 * \boldsymbol{y}, \ \forall \ell \tag{4}$$

$$\boldsymbol{x}^0 = \boldsymbol{W}^0 * \boldsymbol{x}. \tag{5}$$

All other architecture details are kept identical to the ones presented in (Van Den Oord et al., 2016) and (Arik et al., 2017) as best as we could determine. The differences between the two architectures and SynthNet are summarized in Table 1. We compare the performance and quality of these two baselines against SynthNet initially in Table 3 over three sets of hyperparmeters (Table 2). For the best resulting models we perform MOS listening tests, shown in Table 5. Preliminary results for global conditioning experiments are also provided in Table 4.

Table 1: Differences between the two baseline architectures and SynthNet.

|  | Input | | | Dilated conv | Skip | | Final block |
|---|---|---|---|---|---|---|---|
|  | Channels | Type | Activation | Separable | Connection | 1x1 Conv | Activation |
| WaveNet | 1 Scalar | Conv | None | No | Yes | No | ReLU |
| DeepVoice | 256 1-hot | Conv | Tanh | No | Yes | Yes | ReLU |
| SynthNet | 1 Scalar | Embed | Tanh | Yes | No | No | SeLU |

### 3.2 GRAM MATRIX PROJECTIONS

We perform a set of initial experiments to gain more insight towards the learned representations. We use Gram matrices to extract statistics since these have been previously used for artistic style transfer (Gatys et al., 2015). After training, the validation data is fed through five locally conditioned networks, each trained with a distinct harmonic style. The data has identical *melodic* content but has different *harmonic* content (i.e. same song, different instruments). The Gram matrices are extracted from the outputs of each dilated block (Equation 3) for each network - timbre.

These are flattened and projected onto 2D, simultaneously over all layers and styles via T-SNE (Maaten & Hinton, 2008). The results presented in Figure 1 show that the extracted statistics separate further as the layer index increases. A broad interpretation is that the initial layers extract low-level generic audio features, these being common to all waveforms. However, since this is a controlled experiment, we can be more specific. The timbre of a musical instrument is characterized by a specific set of resonating frequencies on top of the fundamental frequency (pure sine wave). Typically one identifies individual notes based on their fundamental, or lowest prominent frequency. These depend on the physics of the musical instrument and effects generated, for example, by the environment. Since the sequence of notes is identical and the harmonic styles differ, we conjectured that Figure 1 could

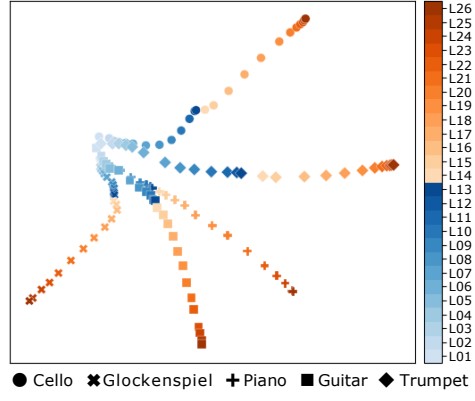

Figure 1: Gram matrix projection from Eq. 3 Layers in color, shapes are styles (timbre).

imply a frequency-layer correspondence. While the latter statement might be loose, the lower layers' statistics are nevertheless much closer due the increased similarity with the fundamental frequency.

## 3.3 SYNTHNET ARCHITECTURE

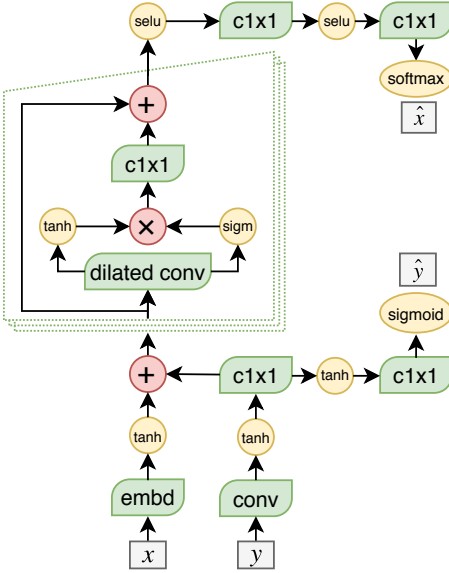

Figure 2: SynthNet (also see Table 1) with a multi-label cross-entropy loss for binary midi.

Figure 1 provides indicative results from many experiments. Additional Gram matrix projections are provided in Appendix B. We hypothesize that the skip connections are superfluous and the conditioning of the first input layer should suffice to drive the melodic component. We also hypothesize that the first audio input layer learns an embedding corresponding to the fundamental pitches. Then, we aim to learn mapping from the symbolic representation (binary midi code) to the pitch embeddings (Equation 8). Therefore, in SynthNet there are no parameters learned in each dilation block for local conditioning and the hidden activation with global conditioning (omitted in Figure 2) becomes

$$h^\ell = \tau\Big(W_f^\ell * x^{\ell-1} + U_f^\ell \cdot z\Big) \odot \sigma\Big(W_g^\ell * x^{\ell-1} + U_g^\ell \cdot z\Big). \tag{6}$$

The input to the dilated blocks is the sum of the embedding codes and the autoencoder latent codes:

$$y^h = \tau(V^0 \cdot y) \tag{7}$$

$$x^0 = \tau(W^0 \cdot x) + \tau(V^h \cdot y^h) \tag{8}$$

$$\hat{y} = V^{out} \cdot \tau(V^h \cdot y^h). \tag{9}$$

As it can be seen in Figure 2 there are no skip connections and Equation 4 no longer applies. We also found that using SeLU activations (Klambauer et al., 2017) in the last layers improves generation stability and quality. Other normalization strategies could have been used, we found SeLU to work well. In addition, we further increase sparsity by changing the dilated convolution in Equation 6 with a dilated depthwise separable convolution. Separable convolutions perform a channel-wise spatial convolution that is followed by a $1 \times 1$ convolution. In our case each input channel is convolved with its own set of filters. Depthwise separable convolutions have been successfully used in mobile and embedded applications (Howard et al., 2017) and in the Xception architecture (Chollet, 2017). As we show in Table 4 and Table 5, the parsimonious approach works very well since it reduces the complexity of the architecture and speeds up training.

In training, SynthNet models the midi data $y$ in an auto-regressive fashion which is similar to the way audio data is modeled, but with a simplified architecture (see Figure 2). In principle, this allows

the model to jointly generate both audio and midi. In practice, the midi part of the model is too simple to generate interesting results in this way. Nonetheless, we found it beneficial to retain the midi loss term during training, which it turns out, tends to act as a useful regularizer — we conjecture by forcing basic midi features to be extracted. In summary, in contrast with Equation 1 we optimize the joint $\log p(\boldsymbol{x}, \boldsymbol{y}|\boldsymbol{z})$, so that

$$\mathcal{L} = -\frac{1}{N}\sum_{i=1}^{N}\left[\sum_{j=1}^{|\boldsymbol{x}|=256} x_j^i \log \hat{x}_j^i + \sum_{j=1}^{|\boldsymbol{y}|=128}\left(y_j^i \log \hat{y}_j^i + (1 - y_j^i)\log(1 - \hat{y}_j^i)\right)\right].$$

## 4 EXPERIMENTS

We compare exact replicas of the architectures described in Van Den Oord et al. (2016); Arik et al. (2017) with our proposed architecture SynthNet. We train the networks to learn the harmonic audio style (here instrument timbre) using *raw audio waveforms*. The network is conditioned locally with a 128 binary vector indicating note on-off, extracted from the *midi files*. The latter describes the melodic content. For the purpose of validating our hypothesis, we decided to eliminate extra possible sources of error and manually upsampled the midi files using linear interpolation. For the results in Table 4 the network is also conditioned globally with a one-hot vector which designates the style (instrument) identity. Hence, multiple instrument synthesizers are learned in a single model. For the hyperparameter search experiments (Table 3) and the final MOS results (Table 5) we train one network for each style, since it is faster.

We use the Adam Kingma & Ba (2014) optimization algorithm with a batch size of 1, a learning rate of $10^{-3}$, $\beta_1 = 0.9$, $\beta_2 = 0.999$ and $\varepsilon = 10^{-8}$ with a weight decay of $10^{-5}$. We find that for most instruments 100-150 epochs is enough for generating high quality audio, however we keep training up to 200 epochs to observe any unexpected behaviour or overfitting. All networks are trained on Tesla P100-SXM2 GPUs with 16GB of memory.

### 4.1 SYNTHETIC REGISTERED AUDIO

We generate the dataset using the freely available Timidity++[1] software synthesizer. For training we selected parts 2 to 6 from Bach's Cello Suite No. 1 in G major (BWV 1007). We found that this was enough to learn the mapping from midi to audio and to capture the harmonic properties of the musical instruments. From this suite, the Prelude (since it is most commonly known) is not seen during training and is used for measuring the validation loss and for conditioning the generated audio.

After synthesizing the audio, we have approximately 12 minutes of audio for each harmonic style, out of which 9 minutes (75%) training data and 3 minutes (25%) of validation data. We experiment with $S = 7$ harmonic styles which were selected to be as different as possible. Each style corresponds to a specific preset from the 'Fluid-R3-GM' sound font. These are (preset number - instrument): S01 - Bright Yamaha Grand, S09 - Glockenspiel, S24 - Nylon String Guitar, S42 - Cello, S56 - Trumpet, S75 - Pan Flute and S80 - Square Lead.

For training, the single channel waveforms are sampled at 16kHz and the bit-depth is reduced to 8 bit via mu-law encoding. Before reducing the audio bit depth, *the waveforms are dithered* using a triangular noise distribution with limits $(-0.009, 0.009)$ and mode 0, which reduces perceptual noise but more importantly keeps the quantization noise out of the signal frequencies. We have found this critical for the learning process.

Without dithering there are melodic discontinuities and clipping errors in the generated waveforms. The latter errors are most likely due to notes getting mapped to the wrong set of frequencies (artifacts appear due to the quantization error). From all harmonic styles, the added white noise due to dithering is most noticeable for Glockenspiel, Cello and Pan Flute. The midi is upsampled to 16kHz to match the audio sampling rate via linear interpolation. Each frame contains a 128 valued vector which designates note on-off times for each note (piano roll).

---

[1]http://timidity.sourceforge.net/

## 4.2 MEASURING AUDIO GENERATION QUALITY

Quantifying the performance of generative models is not a trivial task. Similarly to Van Den Oord et al. (2016); Arik et al. (2017) we have found that once the training and validation losses go beyond a certain lower threshold, the quality improves. However, the losses are only informative towards convergence and overfitting (Figure 3) - they are not sensitive enough to accurately quantify the quality of the generated audio. This is critical for ablation studies where precision is important. Theis et al. (2015) argue that generative models should be evaluated directly. Then, the first option is the mean opinion score (MOS) via direct listening tests. This can be impractical, slowing down the hyperparameter selection procedure. MOS ratings for the best found models are given in Table 5.

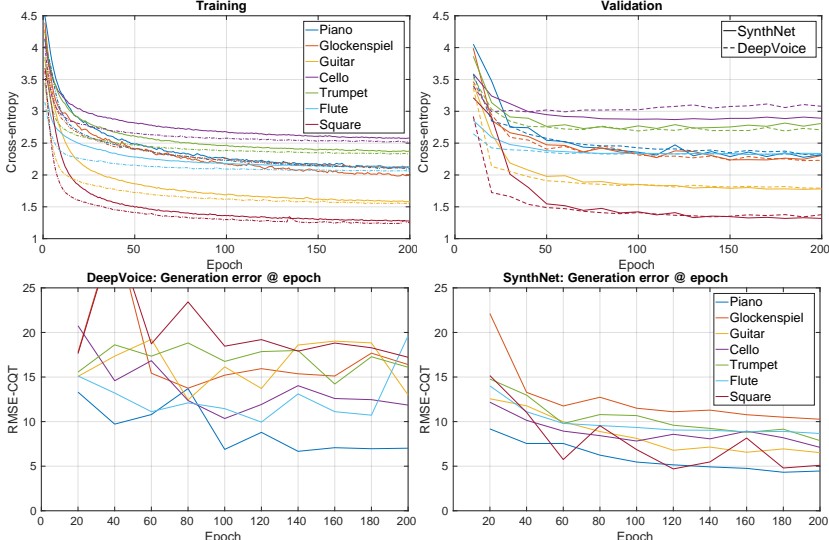

Figure 3: Seven networks are trained, each with a different harmonic style. Top, losses: training (left) validation (right). Bottom, RMSE-CQT: DeepVoice (left [Tbl. 3, col. 6]) and SynthNet (right [Tbl. 3, col. 8]). DeepVoice overfits for Glockenspiel (top right, dotted line). Convergence rate is measured via the RMSE-CQT, not the losses. The capacity of DeepVoice is larger, so the losses are steeper.

Instead, we propose to measure the root mean squared error (RMSE) of the Constant-Q Transform (RMSE-CQT) between the generated audio and the ground truth waveform (Figure 3, lower plots). Similarly to the Fourier transform, the CQT (Brown, 1991) is built on a bank of filters, however unlike the former it has geometrically spaced center frequencies that correspond to musical notes.

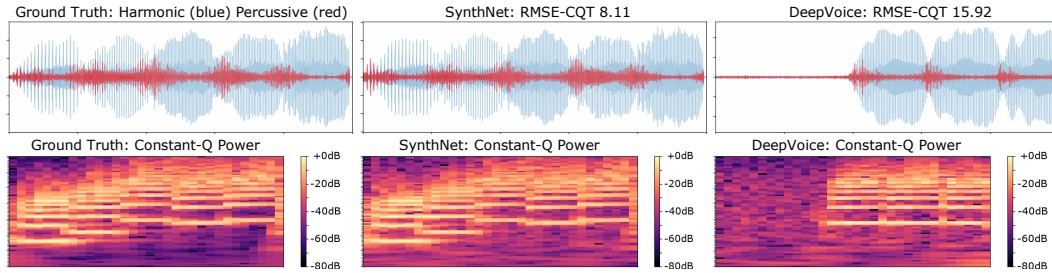

Figure 4: Left: 1 second of ground truth audio of Bach's BWV1007 Prelude, played with FluidSynth preset 56 Trumpet. Center: SynthNet high quality generated. Right: DeepVoice low quality generated showing delay. Further comparisons over other instrument presets are provided in Appendix A. We encourage the readers to listen to the samples here: `http://bit.ly/synthnet_appendix_a`

Other metrics were evaluated as well however only the RMSE-CQT was correlated with the quality of the generated audio. This (subjective) observation was initially made by listening to the audio samples and by comparing the plots of the audio waveforms (Figure 4). Roughly speaking, as we

also show in Figure 4 (top captions) and Figure 3 (lower plots), we find that a RMSE-CQT value below 10 corresponds to a generated sample of reasonable quality. The RMSE-CQT also penalizes temporal delays (Figure 4 - right) and is also correlated with the MOS (Table 3 and Table 5).

We generate every 20 epochs during training and compute the RMSE-CQT to check generation quality. Indeed, Figure 3 shows that the generated signals match the target audio better as the training progresses, while the losses flatten. However, occasionally the generated signals are shifted or the melody is slightly inaccurate - the wrong note is played (Figure 4 - right). This is not necessarily only a function of the network weight state since the generation process is stochastic. We set a fixed random seed at generation time, thus we only observe changes in the generated signal due to weight changes. To quantify error for one model, the RMSE-CQT is averaged over all epochs.

## 4.3 Hyperparameter selection

There are many possible configurations when it comes to the filter width $F$, the number of blocks $B$, and the maximum dilation rate $R$. The dilation rates per each block are: $\{2^0, 2^1, \ldots, 2^{R-1}\}$. In addition there is the choice of the number of residual and skip channels. For speech Arik et al. (2017) use 64 residual channels and 256 skip channels, Engel et al. (2017) use 512 residual channels and 256 skip channels, while Mor et al. (2018) use 512 for both. These methods have receptive field $\Delta < 1$. Since the latter two works are also focused on music, we use 512 channels for both the residual and skip convolutions and set the final two convolutions to 512 and 384 channels respectively.

We hypothesize that it is better to maximize the receptive field $\Delta$ while minimizing the number of layers. Therefore, in our first experiments (Table 3) we limit the receptive field to 1 second and vary the other parameters according to Table 2. We have observed that the networks train faster and the quality is better when the length of the audio slice is maximized within GPU memory constraints.

Table 2: Three setups for filter, dilation and number of blocks resulting in a similar receptive field.

|  | Filter width $F$ | Num blocks $B$ | Max dilation $R$ | Receptive field $\Delta$ |
|---|---|---|---|---|
| **L24** | 3 | 2 | 12 | 1.0239 sec |
| **L26** | 2 | 2 | 13 | 1.0240 sec |
| **L48** | 2 | 4 | 12 | 1.0238 sec |

Table 3: Mean RMSE-CQT and 95% confidence intervals (CIs). Two baselines are benchmarked for three sets of model hyperparameter settings (Table 2), all other parameters identical. One second of audio is generated every 20 epochs (over 200 epochs) and the error versus the target audio is measured and averaged over the epochs, per instrument. Total number of parameters and training time are also given. All waveforms and plots available here: `http://bit.ly/synthnet_table3`

|  | WaveNet | | | DeepVoice | | | SynthNet | | |
|---|---|---|---|---|---|---|---|---|---|
|  | L24 | L26 | L48 | L24 | L26 | L48 | L24 | L26 | L48 |
| S01 | 16.56±4.67 | 14.83±5.39 | 18.15±2.88 | 10.80±1.66 | 9.32±1.97 | 17.28±2.19 | 6.30±1.01 | 5.96±1.10 | *5.51±0.85* |
| S09 | 24.01±5.38 | 22.20±4.56 | 25.47±3.96 | 22.65±5.25 | 17.54±3.86 | 27.48±2.55 | 11.58±1.50 | 12.53±2.37 | *10.91±1.40* |
| S24 | 17.68±3.05 | 18.95±4.13 | 19.30±1.26 | 18.03±1.58 | 16.33±1.79 | 19.19±1.25 | 8.00±1.71 | 8.53±1.51 | *7.82±1.32* |
| S42 | 15.83±3.91 | 17.20±3.53 | 16.29±3.38 | 11.92±0.94 | 13.77±2.06 | 13.89±1.73 | 8.61±1.12 | 8.84±0.96 | *8.33±0.74* |
| S56 | 18.50±2.23 | 17.25±2.98 | 22.89±1.73 | 17.04±0.34 | 17.16±1.05 | 21.45±1.37 | *8.90±0.95* | 10.37±1.41 | 8.97±1.42 |
| S75 | 20.89±6.90 | 20.03±6.73 | 19.78±5.15 | 11.93±1.30 | 12.75±1.93 | 11.30±0.64 | *9.68±1.22* | 9.83±1.10 | 10.20±1.60 |
| S80 | 27.73±2.29 | 26.74±3.92 | 26.96±4.71 | 20.41±1.80 | 20.09±2.91 | 20.95±2.77 | *5.14±1.46* | 7.66±2.31 | 7.91±2.41 |
| All | 20.02±1.79 | **19.60±1.73** | 21.35±1.48 | 16.18±1.30 | **15.31±1.10** | 18.57±1.36 | **8.32±0.63** | 9.10±0.70 | 8.52±0.62 |
| Params | 8.23e+7 | 6.18e+7 | 1.14e+8 | 8.90e+7 | 6.89e+7 | 1.27e+8 | 7.35e+6 | 7.80e+6 | 1.36e+7 |
| Time | 4d2h | 4d2h | 8+ days | 4d10h | 3d9h | 8+ days | 16 hours | 16 hours | 1d5h |

It can be seen in Table 3 that SynthNet outperforms both baselines. Some instruments are more difficult to learn than others (also see Figure 3). This is also observable from listening to and visualizing the generated data (available here `http://bit.ly/synthnet_table3`).

The lowest errors for the first four instruments are observed for SynthNet L48 while the last three are lowest for SynthNet L24 (Table 3 slanted). This could be due to either an increased granularity over the frequency spectrum, provided by the extra layers of the L48 model or a better overlap. The

best overall configuration is SynthNet L24. For DeepVoice and WaveNet, both L24 and L48 have more parameters (Table 3, second last row) and are slower to train, even though all setups have the same number of hidden channels (512) over both baseline architectures. This is because of the skip connections and associated convolutions.

**Global conditioning** We benchmark only DeepVoice L26 against SynthNet L24, with the difference that one model is trained to learn all 7 harmonic styles simultaneously (Table 4). This slows down training considerably. The errors are higher as opposed to learning one model per instrument, however SynthNet has the lowest error. We believe that increasing the number of residual channels would have resulted in lower error for both algorithms. We plan to explore this in future work.

Table 4: RMSE-CQT Mean and 95% CIs. All networks learn 7 harmonic styles simultaneously.

| Experiment | Piano | Glockenspiel | Guitar | Cello | Trumpet | Flute | Square | All | Time |
|---|---|---|---|---|---|---|---|---|---|
| DeepVoice L26 | 14.01±1.41 | 19.68±3.29 | 16.10±1.60 | 13.80±2.11 | 18.68±2.04 | 15.40±3.22 | 15.64±1.76 | 16.19±0.91 | 12d3h |
| SynthNet L26 | 9.37±0.71 | 15.12±3.39 | 11.88±0.95 | 11.66±2.35 | 13.98±1.70 | 12.01±1.36 | 10.90±1.17 | **12.13±0.74** | 5d23h |

## 4.4 MOS LISTENING TESTS

Given the results from Table 3, we benchmark the best performing setups: WaveNet L26, DeepVoice L26 and SynthNet L24. For these experiments, we generate samples from multiple songs using the converged models from all instruments. We generate 5 seconds of audio from Bach's Cello suites not seen during training, namely Part 1 of Suite No. 1 in G major (BWV 1007), Part 1 of Suite No. 2 in D minor (BWV 1008) and Part 1 of Suite No. 3 in C major (BWV 1009) which cover a broad range of notes and rhythm variations.

Table 5 shows that the samples generated by SynthNet are rated to be almost twice as better than the baselines, over all harmonic styles. By listening to the samples (`http://bit.ly/synthnet_mostest`), one can observe that Piano is the best overall learned model, while the basline algorithms have trouble playing the correct melody over longer timespans for other styles. We would also like to remind the reader that all networks have been trained with only 9 minutes of data.

Table 5: Listening MOS and 95% CIs. 5 seconds of audio are generated from 3 musical pieces (Bach's BWV 1007, 1008 and 1009), over 7 instruments for the best found models. Subjects are asked to listen to the ground truth reference, then rate samples from all 3 algorithms simultaneously. 20 ratings are collected for each file. Audio and plots here: `http://bit.ly/synthnet_mostest`

| Experiment | Piano | Glockenspiel | Guitar | Cello | Trumpet | Flute | Square | All |
|---|---|---|---|---|---|---|---|---|
| WaveNet L26 | 2.22±0.25 | 2.48±0.23 | 2.18±0.25 | 2.37±0.28 | 2.18±0.29 | 2.37±0.22 | 2.30±0.09 | 2.30±0.10 |
| DeepVoice L26 | 2.55±0.32 | 1.85±0.23 | 2.30±0.39 | 2.62±0.27 | 2.28±0.32 | 2.20±0.25 | 1.87±0.03 | 2.24±0.11 |
| SynthNet L24 | 4.75±0.14 | 4.45±0.17 | 4.30±0.19 | 4.50±0.15 | 4.25±0.18 | 4.15±0.21 | 4.10±0.16 | **4.36±0.07** |

## 5 DISCUSSION

In the current work we gave some insights into the learned representations of generative convolutional models. We tested the hypothesis that the first causal layer learns fundamental frequencies. We validated this empirically, arriving at the SynthNet architecture which converges faster and produces higher quality audio.

Our method is able to simultaneously learn the characteristic harmonics of a musical instrument (timbre) and a joint embedding between notes and the corresponding fundamental frequencies. While we focus on music, we believe that SynthNet can also be successfully used for other time series problems. We plan to investigate this in future work.

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

# A  GENERATED WAVEFORMS VS. GROUND TRUTH

Audio samples and visualizations here: `http://bit.ly/synthnet_appendix_a`

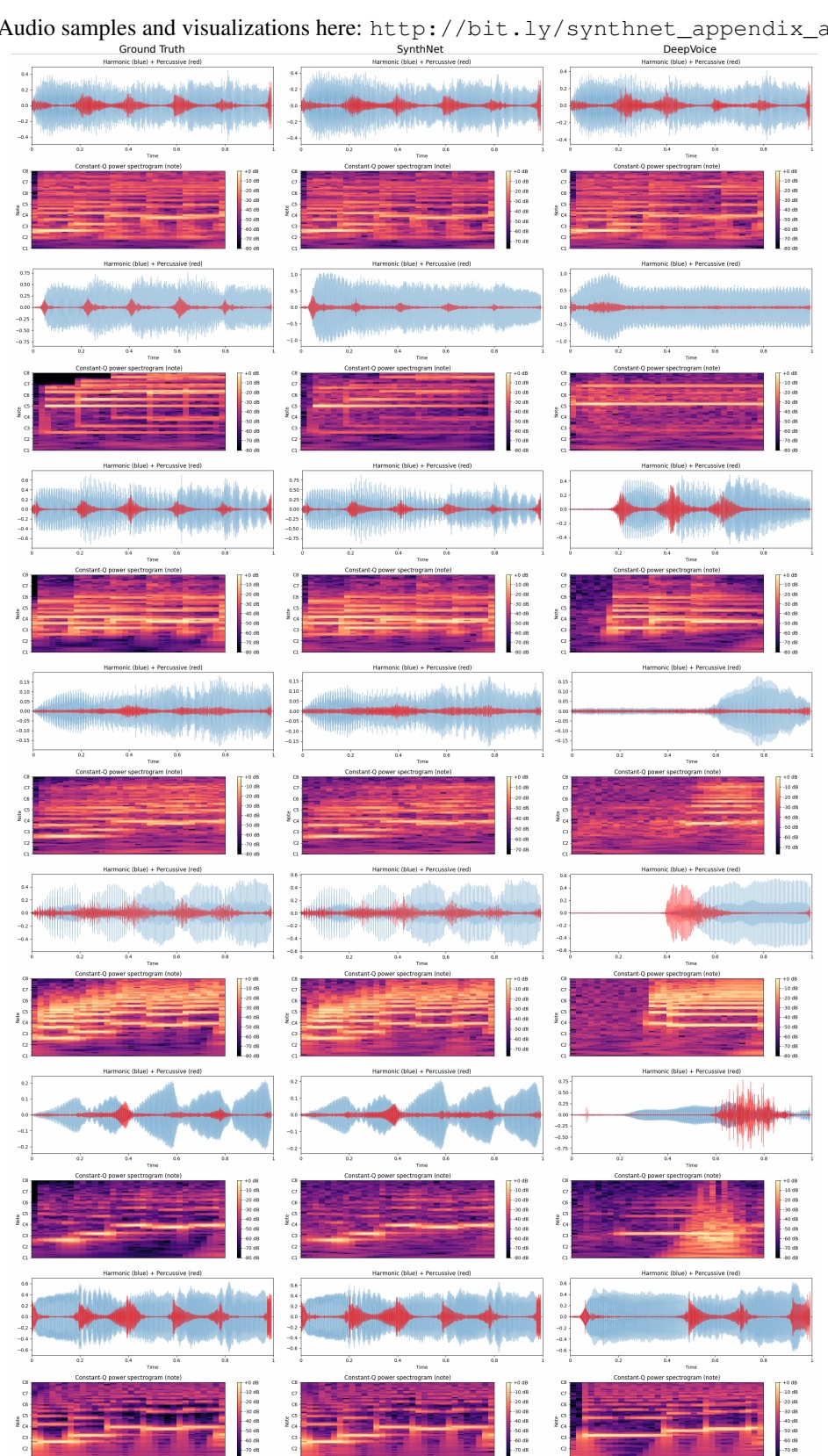

# B  GRAM MATRIX PROJECTIONS

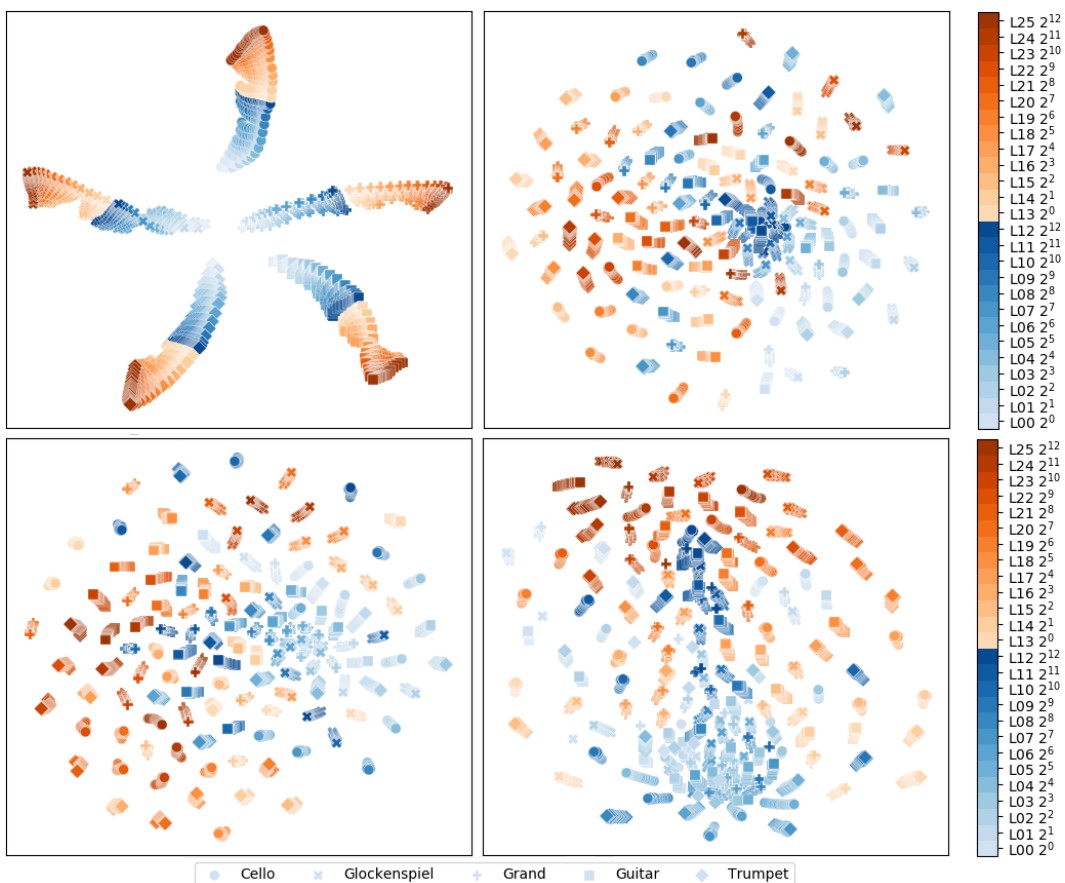

Figure 5: Gram matrices extracted during training, every 20 epochs. Top left: extracted from Equation 3. Top right: extracted from Equation 2. Bottom left: extracted from the filter part of Equation 2. Bottom right: extracted from the gate part of Equation 2.

