# OpenReview forum: "Synthnet: Learning synthesizers end-to-end"
_ICLR.cc/2019/Conference_

### Official Review · AnonReviewer2 · 2018-11-02
**use of synthetic dataset makes the work much less impactful**

**Rating:** 3
**Confidence:** 5

**Review:**

This paper proposes a neural model for synthesizing instrument sounds, using an architecture based on the WaveNet and DeepVoice models. The model generates raw waveforms conditioned on a piano roll representation of aligned MIDI input.

My biggest gripe with this work is that the model is trained entirely on a synthetic dataset generated from a sample-based synthesizer using a sound font. I feel that this defeats the purpose, as it will never work better than just sampling the original sound library. One potential argument in favour would be to save storage space, but the sound font used for the work is only ~140 MB, which is not prohibitive these days (indeed, many neural models require a comparable amount of storage).

It would be much more interesting to train the model on real instrument recordings, because then it could capture all the nuances of the instruments that sample-based synthesizers cannot replicate. As it stands, all the model has to do is reproduce a fixed (and fairly small) set of audio samples. This is arguably a much simpler task, which could also explain why reducing the model size (SynthNet's depthwise convolutions have many fewer parameters than the regular convolutions used in WaveNet and DeepVoice) works so well here.

That said, I think the proposed architectural modifications for raw audio models could be interesting and should be tested for other, more challenging tasks. The proposed RMSE-CQT error measure is potentially quite valuable for music generation research, and its correlation with MOS scores is promising (but this should also be tested on more realistic audio).

The fact that the models were trained to convergence on only 9 minutes of data per instrument is also impressive, despite the limitations of the dataset. The use of dithering to reduce perceptual noise is also interesting and some comparison experiments there would have been interesting, especially to corroborate the claim that it is critical for the learning process.

I think the paper slightly overstates its contributions in terms of providing insight into the representations that are learned in generative convolutional models. The Gram matrix projections showing that the activations of different layers diverge for different input types as we advance through the model is not particularly surprising, and similar plots could probably be made for almost any residual model.

Overall, I feel the work has some fundamental flaws, mostly stemming from the dataset that was used.



Miscellany:

- In the abstract: "is substantially better in quality", compared to what?

- In the introduction, it is mentioned that words in a speech signal cannot overlap, but notes in a musical signal can. I would argue that these are not comparable abstractions though, words themselves are composed of a sequence of phonemes, which are probably a better point of comparison (and phonemes, while they don't tend to overlap, can affect neighbouring phonemes in various ways). That said, I appreciate that this is probably quite subjective.

- Overall, the formulation of paragraph 2 of the introduction is a bit unusual, I think the same things are said in a much better way in Section 3.

- "Conditioning Deep Generative Raw Audio Models for Structured Automatic Music" by Manzelli et al. (2018) also proposes a MIDI-conditional neural audio generation model, trained on real instrument recordings from the MusicNet dataset. I think this is a very relevant reference.

- In the contributions of the paper, it is stated that "the generated audio is practically identical to ground truth as can be seen in Figure 4" but the CQTs in this figure are visibly different.

- I don't think it is fair to directly compare this setup to Engel et al. (2017) and Mor et al. (2018) as is done in the last paragraph of Section 2, as these are simply different tasks (mapping from audio to audio as opposed to generating audio).

- At the start of Section 3.1 it would be good to explicitly mention whether 8-bit mu-law audio is used, to explain why the waveform is 256-valued.

- Why is the conditioning causal? It does not need to be, as the piano roll is fully available in advance of the audio generation. I guess one argument in favour would be to enable real-time generation, but it would still be good to compare causal and non-causal conditioning.

- Since the piano roll representation is binary, does that mean MIDI velocity is not captured in the conditioning signal? It would probably be useful for the model to provide this information, so it can capture the differences in timbre between different velocities.

- The use of a MIDI prediction loss to regularise the conditioning part of the model is interesting, but I would have liked to see a comparison experiment (with/without).

- In Section 4.3, specify the unit, i.e. "Delta < 1 second".

- For the task of recreating synthetic audio samples, the WaveNet models seem to be quite large. As far as I can tell the size hyperparameters were chosen based on the literature, but the inherited parameters were originally optimised for different tasks.

- In Section 4.3 under "global conditioning", the benchmark is said to be between DeepVoice L26 and SynthNet L24, but Table 4 lists DeepVoice L26 and SynthNet L26, which version was actually used?

---

> ### Author Response · Authors · 2018-11-21
> **In relation to your minor comments 2**
>
> "- In the contributions of the paper, it is stated that "the generated audio is practically identical to ground truth as can be seen in Figure 4" but the CQTs in this figure are visibly different."
>
>  Thank you for your comment. Indeed, there are minute differences. The CQTs are quite similar and very accurate, especially when compared to the baselines, as it can also be seen in Appendix 1. To the best of our knowledge this is the first time that such accurate signals are generated and measured directly vs the ground truth.
>
>
> "- I don't think it is fair to directly compare this setup to Engel et al. (2017) and Mor et al. (2018) as is done in the last paragraph of Section 2, as these are simply different tasks (mapping from audio to audio as opposed to generating audio)."
>
> The architectural components in Engel et al. (2017) and Mor et al. (2018) are almost identical to the original work of Van Den Oord et al. (2016) with the difference that a separate simpler residual network is used for the encoder.  In our work, we learn an auxiliary task based on the midi data instead of a separate encoder network and this is how the tasks are different.
>
>
> "- At the start of Section 3.1 it would be good to explicitly mention whether 8-bit mu-law audio is used, to explain why the waveform is 256-valued."
>
>  Thank you for your comment.
>
>
> "- Why is the conditioning causal? It does not need to be, as the piano roll is fully available in advance of the audio generation. I guess one argument in favour would be to enable real-time generation, but it would still be good to compare causal and non-causal conditioning."
>
>  Thank you for your comment. Indeed, it is necessary to restrict to a causal model if we are to synthesise in real time (i.e. use it to generate sound from a midi keyboard). It would be interesting to consider non-causal models but it is somewhat orthogonal to the main aim of our work.
>
>
> "- Since the piano roll representation is binary, does that mean MIDI velocity is not captured in the conditioning signal? It would probably be useful for the model to provide this information, so it can capture the differences in timbre between different velocities."
>
>  Thank you for your comment. Indeed, we do not use velocity information and doing so can improve expressiveness.
>
>
> "- The use of a MIDI prediction loss to regularise the conditioning part of the model is interesting, but I would have liked to see a comparison experiment (with/without)."
>
> Thank you for your comment. Again, we were limited by the page limit. The additional task helps with learning a joint representation.
>
>
> " - In Section 4.3, specify the unit, i.e. "Delta < 1 second". "
>
>  Thank you for your comment.
>
>
> " - For the task of recreating synthetic audio samples, the WaveNet models seem to be quite large. As far as I can tell the size hyperparameters were chosen based on the literature, but the inherited parameters were originally optimised for different tasks. "
>
> Thank you for your comment. The tasks in Engel et al. (2017) and Mor et al. (2018) are similar as mentioned before. Learning the timbre (harmonic series) is correlated with model size.
>
>
> "- In Section 4.3 under "global conditioning", the benchmark is said to be between DeepVoice L26 and SynthNet L24, but Table 4 lists DeepVoice L26 and SynthNet L26, which version was actually used?"
>
> Thank you for your comment. We will update the table to L24.

---

> ### Author Response · Authors · 2018-11-21
> **In relation to your minor comments 1**
>
> "Miscellany:
> - In the abstract: "is substantially better in quality", compared to what?"
>
> SynthNet is compared with two baselines, WaveNet and DeepVoice. Quality includes reproducing the timbre (easier) as well as the melody (difficult).
>
> "- In the introduction, it is mentioned that words in a speech signal cannot overlap, but notes in a musical signal can. I would argue that these are not comparable abstractions though, words themselves are composed of a sequence of phonemes, which are probably a better point of comparison (and phonemes, while they don't tend to overlap, can affect neighbouring phonemes in various ways). That said, I appreciate that this is probably quite subjective."
>
> Thank you for your comment. Phonemes are indeed less complex than the harmonics of any instrument, even synthetic (see overtone series). While phonemes do not overlap, notes do.
>
>
> "- Overall, the formulation of paragraph 2 of the introduction is a bit unusual, I think the same things are said in a much better way in Section 3."
>
> Thank you for your comment.
>
>
> " - "Conditioning Deep Generative Raw Audio Models for Structured Automatic Music" by Manzelli et al. (2018) also proposes a MIDI-conditional neural audio generation model, trained on real instrument recordings from the MusicNet dataset. I think this is a very relevant reference."
>
> The authors use subsets (solo performances) of the MusicNet and YouTube dataset to fine tune and generate novel music with a better learned compositional (melodic) structure via LSTMs. In a nutshell, a WaveNet is locally conditioned on LSTM outputs (the latter trained on MIDI data). The WaveNet and LSTMs are trained separately.
>
> When it comes to the ability of the WaveNet in the mentioned article to learn synthesisers, the evaluation method in the article is debatable (no MOS tests / measurements are provided) and the ambiguity arises from the objective itself: generating new compositions.
>
> Even with a direct measurement of error (which is not provided), it is not possible to know whether the errors (versus a ground truth) in the generated melodies arise from the WaveNet (undesirable) or whether they are novel compositions / variations based on the LSTM (desirable).
>
> The authors mention in a paragraph that they were able to generate the C scale (a simple sequence of 16 MIDI notes, 8 of which are unique) and a short part of Happy Birthday (a sequence of 12 MIDI notes, 7 of which are unique) using a locally conditioned WaveNet trained on the solo cello subset of the MusicNet dataset. The quality of the generated samples is poor and by far not comparable with the melodic and harmonic fidelity of SynthNet. Not all notes from the C scale are present, there are broad volume / velocity variations (generation errors). The quality and accuracy is not measured using MOS tests and no direct comparisons are provided either.
> Having said that, the two samples are comparable to what the baselines in our work can produce on more complex melodies. By induction, this essentially validates the SynthNet architecture (which is proven better than the baselines) for real instrument recordings.
> In our work, we provide a thorough evaluation of the baselines and SynthNet with far more complex melodies and over 7 timbres.
>
> Furthermore, it is interesting why the next section in the article is focused on novel compositions (subjective) instead of focusing on accurate generation (measureable, via MOS tests and as we show – via direct comparisons).
>
> Moreover, the usability of MusicNet for learning music synthesisers is debatable. This stems from the varying quality of the audio, the recording conditions / environment and the alignment quality. MusicNet doesn’t contain that many solo recordings (e.g. 15 hours of solo piano, 30 minutes of solo violin and 49 minutes of solo cello). All the latter are not of the same quality / consistency.
>  The varying recording conditions in MusicNet might be desirable for classification, timbre recognition or other tasks, however this is not desirable for learning synthesisers where a dataset containing consistent recordings is a more sensible choice.

---

> ### Author Response · Authors · 2018-11-21
> **In relation to your sixth comment**
>
> "Overall, I feel the work has some fundamental flaws, mostly stemming from the dataset that was used."
>
> Thank you for your comments. We were limited by the number of pages and could not include more experimental results. We initially used one hour of training data, however we did not see any of the models doing worse when trained with less data. Running experiments faster is advantageous while searching for hyperparameters that worked well over all 7 timbres.
> Of course, for an even more varied set of test songs (other composers, genres etc) more training data would be needed, but as we previously argue, good quality data is hard to come by.

---

> ### Author Response · Authors · 2018-11-21
> **In relation to your fifth comment**
>
> "I think the paper slightly overstates its contributions in terms of providing insight into the representations that are learned in generative convolutional models. The Gram matrix projections showing that the activations of different layers diverge for different input types as we advance through the model is not particularly surprising, and similar plots could probably be made for almost any residual model."
>
> This is indeed true in general (Gram matrix projection), as we also mentioned in the article. However, the structured nature of music and instrument timbre (overtone series) allows us to draw more specific conclusions. Thinking in this setting led to the use of the auxiliary training task and learning the joint representation between fundamental frequency and piano roll data which effectively enables the accurate, controlled generation of signals, in this case music, in other words, learn synthesisers end to end.
>
> By better understanding the learned representations, hypothesising and experimenting, we arrived at the SynthNet architecture which is able to accurately learn virtual instruments and generate high fidelity and accurate waveforms, without the need for individually labelled notes ( Van Den Oord et al. (2016) do not use individually labelled phonemes).

---

> ### Author Response · Authors · 2018-11-21
> **In relation to your fourth comment**
>
> "The fact that the models were trained to convergence on only 9 minutes of data per instrument is also impressive, despite the limitations of the dataset. The use of dithering to reduce perceptual noise is also interesting and some comparison experiments there would have been interesting, especially to corroborate the claim that it is critical for the learning process."
>
> Unfortunately, we were limited by the number of pages and could not add further experimental results. Not using dithering for either the baselines or SynthNet results in melodies that are not accurate (the melody diverges quickly due to the autoregressive process). This happens because the mapping between fundamental frequency and label is corrupted with correlated (overtone) noise which is an artefact of the bit depth reduction (mu-law encoding) process. This is perhaps not so dramatic for speech. This is also clearly mentioned in the article.

---

> > ### Comment · AnonReviewer2 · 2018-11-21
> > **reviewer response**
> >
> > The ICLR enforced page limit is 10 pages, with a recommendation to stick to 8 pages. I would wager that a description of this ablation experiment as well as a table with results would take up about a quarter of a page at most. I don't think space limitations are a valid reason for not including more experiments here, especially given the overlap between the introduction and Section 3 (many of the things mentioned there are basically the same). More space could definitely be freed up by reformulating some things and reducing duplication.

---

> > > ### Author Response · Authors · 2018-11-23
> > > **author response**
> > >
> > > The ICLR enforced page limit is 10 with a strong suggestion to stick to 8 pages.
> > >
> > > Any article can be improved ad infinitum.
> > >
> > > We have made several contributions, set standards and laid some fundamental work.
> > >
> > > We appreciate your suggestions and will consider them as extensions in future work and encourage the community to replicate our work and extend it.

---

> ### Author Response · Authors · 2018-11-21
> **In relation to your third comment**
>
> "That said, I think the proposed architectural modifications for raw audio models could be interesting and should be tested for other, more challenging tasks. The proposed RMSE-CQT error measure is potentially quite valuable for music generation research, and its correlation with MOS scores is promising (but this should also be tested on more realistic audio)."
>
> Thank you for your comment. We demonstrate that it is possible to create instrument synthesisers using a thorough methodology and over extensive tests. We preferred to be in control of the training data and as such used synthesisers to generate training data, thus making sure that there were no alignment errors and the audio quality was similar (recording equipment, acoustics, etc). Similar audio quality is highly realistic for learning high flidelity synthesisers. Labelled recordings of similar quality (and recorded in similar conditions) are very hard to come by.

---

> ### Author Response · Authors · 2018-11-21
> **In relation to your second comment**
>
> "It would be much more interesting to train the model on real instrument recordings, because then it could capture all the nuances of the instruments that sample-based synthesizers cannot replicate. As it stands, all the model has to do is reproduce a fixed (and fairly small) set of audio samples. This is arguably a much simpler task, which could also explain why reducing the model size (SynthNet's depthwise convolutions have many fewer parameters than the regular convolutions used in WaveNet and DeepVoice) works so well here."
>
> We would respectfully like to argue that learning high fidelity synthesisers is not a simple task. We demonstrate that SynthNet can accurately generate up to 5 seconds of audio from 3 melodies, over 7 timbres. There is no reproduction, the 3 generated melodies are not part of the training set. More critically, the length of the generated audio is not crucial in demonstrating melody accuracy, but the number of notes in the melody (e.g. one could play 3 notes very slowly or very fast - resulting in varying audio lengths).
>
> SynthNet can generate accurately over longer time spans, however we limited the generated audio length to 5 seconds for the MOS listening tests to make it easy for the human evaluators to directly compare the melody against the ground truth. Each subject listens to the ground truth, then a random permutation of the baselines and SynthNet. We also compare generated audio directly based on the raw waveforms which are very similar - unlike previous work in speech. Direct measurements have not been provided before for audio.
>  A thorough evaluation using real instrument recordings implies that the recordings would be made under similar conditions (recording equipment, mastering, acoustics, etc) and the alignment would not contain many errors, if any. Such a dataset is not easy to find, so we resorted to generating our own data.
>
> Of course, a model that can learn synthesisers despite differing recording conditions would be desirable, this however would intuitively require much more data and quality alignment (good labels). Moreover, this is somewhat orthogonal to our goal and not the first step in the process of learning synthesisers based on real recordings.

---

> ### Author Response · Authors · 2018-11-21
> **In relation to your first comment**
>
> "My biggest gripe with this work is that the model is trained entirely on a synthetic dataset generated from a sample-based synthesizer using a sound font. I feel that this defeats the purpose, as it will never work better than just sampling the original sound library. One potential argument in favour would be to save storage space, but the sound font used for the work is only ~140 MB, which is not prohibitive these days (indeed, many neural models require a comparable amount of storage)."
>
> Our goal was to demonstrate that it is possible to generate high fidelity harmonically and melodically accurate raw waveforms i.e. learn synthesisers. We demonstrate this using minimal training data and showed that SynthNet trains much faster than the baselines.
> Our evaluation process is thorough. We do this over 7 timbres. The evaluation is done via MOS tests and by directly measuring raw audio – which is novel in itself.
> We do not demonstrate further expressiveness with real data due to the space constraints in the article and scarcity of quality data. However, our method makes increased expressiveness possible, unlike training with a dataset of individually labelled (NSynth Engel et al. 2017).

---

> > ### Comment · AnonReviewer2 · 2018-11-21
> > **reviewer response**
> >
> > Fair enough, but I still see no clear advantages of doing this over simply using the original synthesizer. It is not surprising that these models can learn to reproduce a small set of sequences sample-by-sample fairly accurately -- it is much more interesting to show that they can generalise in interesting ways, e.g. by training on real data.

---

> > > ### Author Response · Authors · 2018-11-23
> > > **author response**
> > >
> > > Our work builds the foundations and can be seen as an extension of Engel et al (2017) eliminating the need for a dataset of individually labelled notes. Our aim is to set a standard for future work on generative models for sequences as we have shown how to measure directly, and to the best of our knowledge this is the first time a generative sequence model is able to generate sequences conditionally with very high fidelity.
> > >
> > > Stating that the models reproduce a sequence of samples is subjective - we have shown that the validation loss does not increase for SynthNet so it is not simply memorizing the training data. In other words, SynthNet can generalize to other sequences (melodies) and timbres as we have shown for 7 different timbres and 3 different melodies.
> > >
> > > The challenge lies in generating a melody accurately and we have shown that SynthNet can do that very well. The baselines (conditioned WaveNet, DeepVoice) can reproduce timbre, however the generated melody (conditioning signal) is inaccurate even though the timbre is similar.
> > >
> > > Perhaps a better way to highlight the main contribution of our work is to make a parallel with other generative models: the baselines are like vanilla GANs - they can generate data from the same distribution, but without much control over what the content is / what gets generated. In contrast, SynthNet is like VAEs where there is an explicit control over what is generated. In summary, the conditioned baslelines can generate sequence data without accurate control of the content (i.e. new melody compositions), while SynthNet can do both.
> > >  We also wonder why the referred work focuses on generating novel compositions when it claims to briefly show that they can learn synthesisers from MusicNet. Why not perfect that work instead? If it would have been easy to learn synthesisers - why not focus in that direction instead of something subjective / unmeasurable?
> > >
> > > Finally, the usefulness of the advertised dataset is debatable for learning synthesisers since the quality of the recordings vary and the recording conditions are inconsistent.

---

### Official Review · AnonReviewer1 · 2018-11-04
**Learning a neural music synthesizer from a software MIDI synthesizer seems hard to do while having a strong ceiling on performance**

**Rating:** 4
**Confidence:** 4

**Review:**

This paper describes the use of a wavenet synthesizer conditioned on a piano-roll representation to synthesize one of seven different instruments playing approximately monophonic melodic lines.  The system is trained on MIDI syntheses rendered by a traditional synthesizer.

While the idea of end-to-end training of musical synthesizers is interesting and timely, this formulation of the problem limits the benefits that such a system could provide.  Specifically, it would be useful for learning expressive performance from real recordings of very expressive instruments.  For example, in the provided training data, the trumpet syntheses used to train this wavenet sound quite unconvincing and unexpressive.  Using real trumpet performances could potentially learn a mapping from notes to expressive performance, including the details of transitions between notes, articulation, dynamics, breath control, etc.  MIDI syntheses have none of these, and so cannot train an expressive model.

While the experiments show that the proposed system can achieve high fidelity synthesis, it seems to be on a very limited sub-set of musical material.  The model doesn't have to learn monophonic lines, but that seems to be what it is applied on.  It is not clear why that is better than training on individual notes, as Engel et al (2017) do.  In addition, it is trained on only 9 minutes of audio, but takes 6 days to do so.  This slow processing is somewhat concerning.  In addition, the 9 minutes of audio seems to be the same pieces played by each instrument, so really it is much less than 9 minutes of musical material.  This may have implications for generalization to new musical situations and contexts.

Overall, this is an interesting idea, and potentially an interesting system, but the experiments do not demonstrate its strengths to the extent that they could.


Minor comments:

* The related work section repeats a good amount of information from the introduction. It could be removed from one of them

* Table 1: I don't understand what this table is describing.  SynthNet is described as having 1 scalar input, but in the previous section said that it had 256-valued encoded audio and 128-valued binary encoded MIDI as input.

* The use of the term "style" to mean "timbre" is confusing throughout and should be corrected.

* Figure 1: I do not see a clear reason why there should be a discontinuity between L13 and L14, so I think it is just a poor choice of colormap.  Please fix this.

* Page 5: MIDI files were upsampled through linear interpolation.  This is a puzzling choice as the piano-roll representation is supposed to be binary.

* Page 7: "(Table 3 slanted)" I would either say "(Table 3, slanted text)" or "(Table 3, italics)".

* Page 8: "are rated to be almost twice as better" this should be re-worded as "twice as good" or something similar.

---

> ### Author Response · Authors · 2018-11-21
> **In relation to your minor comments**
>
> "Minor comments:
> * The related work section repeats a good amount of information from the introduction. It could be removed from one of them "
>
> Thank you for your comment.
>
> "* Table 1: I don't understand what this table is describing. SynthNet is described as having 1 scalar input, but in the previous section said that it had 256-valued encoded audio and 128-valued binary encoded MIDI as input. "
>
> Mu-law encoding reduces the bit depth from 16 bit (2^16 = 65k) to 8 bit (2^8 = 256). So, the input is scalar and can have scalar values ranging from 0 to 255. These can also be encoded as a 1-hot vector, but this is not done for SynthNet. The midi is processed into piano roll where each frame is a binary vector with 128 values.
>
> "* The use of the term "style" to mean "timbre" is confusing throughout and should be corrected. "
>
>  Thank you for your comment.
>
> "* Figure 1: I do not see a clear reason why there should be a discontinuity between L13 and L14, so I think it is just a poor choice of colormap. Please fix this. "
>
> Thank you for your comment. Every block of repeated dilations is coloured differently.
>
> "* Page 5: MIDI files were upsampled through linear interpolation. This is a puzzling choice as the piano-roll representation is supposed to be binary. "
>
> Thank you for your comment. Indeed, the article miswords this - no linear interpolation is needed, rather we forward fill the binary midi data to obtain our temporal grid version at the appropriate sampling rate. During the piano roll data construction, we simply treat the midi note on/offs as points in continuous time and look up whether the note is on at the given grid point, where the grid may be arbitrarily fine grained. One could call this “forward filling” rather than “linear interpolation”.
>
> "* Page 7: "(Table 3 slanted)" I would either say "(Table 3, slanted text)" or "(Table 3, italics)".  "
>
> Thank you for your comment.
>
> "* Page 8: "are rated to be almost twice as better" this should be re-worded as "twice as good" or something similar. "
>
> Thank you for your comment.

---

> ### Author Response · Authors · 2018-11-21
> **In relation to your third comment**
>
> "In addition, it is trained on only 9 minutes of audio, but takes 6 days to do so. This slow processing is somewhat concerning. In addition, the 9 minutes of audio seems to be the same pieces played by each instrument, so really it is much less than 9 minutes of musical material. This may have implications for generalization to new musical situations and contexts. Overall, this is an interesting idea, and potentially an interesting system, but the experiments do not demonstrate its strengths to the extent that they could. "
>
> SynthNet trains in 16 to 30 hours depending on the number of layers while the baselines can take up to 6 days to train and reach the exact same number of epochs while still not generating accurate melodies.
>
> Training can be stopped earlier for all models (say 100 epochs), however we show that unlike the baselines, SynthNet does not overfit even when training continues up to 200 epochs. In summary, 8 to 15 hours is enough time to train SynthNet on a dataset of 9 minutes, which means that it would not take more than 1-2 hours of training time per minute of training data for SynthNet.
> Furthermore, all models, including the baselines are trained on a single GPU card and training is not parallelized or optimized.
>
> Indeed, training on a very limited set of musical material is one of the strengths of SynthNet. Using minimal data (9 minutes of Bach’s 1st Cello suite) the model can learn synthesisers and play Bach’s 2nd and 3rd Cello suites accurately (not seen during training), as demonstrated over 7 musical timbres, while the baselines WaveNet and DeepVoice do not produce accurate melodies (see Figure 4 – right and Appendix 1 - right).
>
> There are 9 minutes of audio for each timbre. For all experiments except for the experiments in Table 4, each model is trained with only its own data (one of the timbres). So, each model is indeed trained with exactly 9 minutes of data.

---

> ### Author Response · Authors · 2018-11-21
> **In relation to your second comment**
>
> "The system is trained on MIDI syntheses rendered by a traditional synthesizer. While the idea of end-to-end training of musical synthesizers is interesting and timely, this formulation of the problem limits the benefits that such a system could provide. Specifically, it would be useful for learning expressive performance from real recordings of very expressive instruments. For example, in the provided training data, the trumpet syntheses used to train this wavenet sound quite unconvincing and unexpressive.
> Using real trumpet performances could potentially learn a mapping from notes to expressive performance, including the details of transitions between notes, articulation, dynamics, breath control, etc. MIDI syntheses have none of these, and so cannot train an expressive model. While the experiments show that the proposed system can achieve high fidelity synthesis, it seems to be on a very limited sub-set of musical material. The model doesn't have to learn monophonic lines, but that seems to be what it is applied on. It is not clear why that is better than training on individual notes, as Engel et al (2017) do. "
>
> Training end to end via aligned waveforms and midi brings forward the possibility of learning more expressive variations as opposed to training on individually labelled notes and our work takes the state of the art closer towards increased expressiveness.
> There are indeed expressive variations such as legato which describe smooth transitions between notes - a dataset of individually labelled notes does not contain such variations.
>
> Thus far, we are not aware of any method that can capture the expressiveness level described by the reviewer. With the current work we move a step closer towards increased expressivity by achieving similar synthesiser results as Engel et al (2017) but without using individually labelled notes and using much less data while training faster.
>
> Recording a dataset of individually labelled notes for a real instrument is very laborious, and it limits the expressiveness level. Conversely, our method offers the possibility of creating more expressive synthesisers since the recording process is easier (i.e. record while an artist is playing several musical pieces using the target instrument).
>
> Regarding the trumpet samples, indeed, they do sound artificial, and that is because the training data is based on virtual instruments, which are artificial. However, it does sound like the virtual trumpet instrument so the harmonic component is learned, just like all the other 6 timbres. More importantly, the melody is accurate and this is a more difficult challenge given the minimal training data.
> Furthermore, the hyperparameter setup is identical over all instruments / timbres. With more tuning, the expressiveness can be improved further, for all timbres.

---

> ### Author Response · Authors · 2018-11-21
> **In reation to your first comment**
>
> "This paper describes the use of a wavenet synthesizer conditioned on a piano-roll representation to synthesize one of seven different instruments playing approximately monophonic melodic lines. "
>
> To the best of our knowledge SynthNet is the first autoregressive convolutional generative model that can generate high resolution sequential data that is nearly identical to the ground truth, based on minimal training data. This is why we are able to directly measure the generation accuracy. We demonstrate this with simple piano roll data and synthesised audio from 7 distinct timbres.
>
> Indeed, more complex melodic and harmonic signals can be used as training data, however we achieve similar results to Engel et al (2017) (their data is also synthetic) with less data, faster training and with the potential for further expressiveness.
>
> In their work, Van Den Oord et al. (2016) revolutionized text to speech. However, while the speech waveforms sound identical, they are very different to the actual ground truth speech.
>
> SynthNet produces waveforms that are very similar to the ground truth (Fig. 4 and Appendix 1) as demonstrated for 7 virtual instruments. Our method overcomes both challenges of capturing the instrument timbre while accurately reproducing the content – the latter being more challenging.

---

### Official Review · AnonReviewer3 · 2018-11-10
**SoundFont-rendered audio is too restricted a domain on which to draw conclusions about music generation**

**Rating:** 4
**Confidence:** 3

**Review:**

This paper proposes several architecture changes to a WaveNet-like dilated convolutional audio model to improve performance for MIDI-conditioned single-instrument polyphonic music generation.

The experimental results and provided samples do clearly show that the proposed architecture does well at reproducing the sounds of the training instruments for new MIDI scores, as measured by CQT error and human preference.  However, the fact that the model is able to nearly-exactly reproduce CQT is contrary to intuition; given only note on/off times, for most instruments there would be many perceptually-distinct performances of those notes.  This suggests that the task is too heavily restricted.

It isn't clearly stated until Section 4 that the goal of the work is to model SoundFont-rendered music.  (The title "SynthNet" is suggestive but any music generated by such an audio model could be considered "synthesized".)  Using a SoundFont instead of "real" musical recordings greatly diminishes the usefulness of this work; adding and concatenating outputs from the single-note model of Engel et al. removes any real need to model polyphony, and there's no compelling argument that the proposed architecture changes should help in other domains.

One change that could potentially increase the paper's impact is to train and evaluate the model on MusicNet (https://homes.cs.washington.edu/~thickstn/musicnet.html), which contains 10+ minutes of recorded audio and aligned note labels for each of ~5 single instruments (as well as many ensembles).  This would provide evidence that the proposed architecture changes improve performance on a more realistic class of polyphonic music.

Another improvement would be to perform an ablation study over the many architecture changes.  This idea is mentioned in 4.2 but seemingly dismissed due to the impracticality of performing listening studies, which motivates the use of RMSE-CQT.  However, no ablation study is actually performed, so it's not obvious what readers of the paper should learn from the new architecture even restricted to the domain of SoundFont-rendered music generation.


Minor points / nitpicks:

One of the claimed contributions is dithering before quantization to 8-bits.  How does this compare to using mixtures of logistics as in Salimans et al. 2017?

S2P3 claims SynthNet does not use note velocity information; this is stated as an advantage but seems to make the task easier while reducing applicability to "real" music.

S4P1 and S4.1P4 state MIDI is upsampled using linear interpolation.  What exactly does this mean?  Also, the representation is pianoroll if I understand correctly, so what does it mean to say that each frame is a 128-valued vector with "note on-off times"?  My guess is it's a standard pianoroll with 0s for inactive notes and 1s for active notes, where onsets and offsets contain linear fades, but this could be explained more clearly.

What is the explanation of the delay in the DeepVoice samples?  If correcting this is just a matter of shifting the conditioning signal, it seems like an unfair comparison.

S1P2 points (2) and (3) arguing why music is more challenging than speech are questionable.  The timbre of a real musical instrument may be more complex than speech, but is this true for SoundFonts where the same samples are used for multiple notes?  It's not clear what the word "semantically" even means with regard to music.

The definition of timbre in S3.2P2 is incomplete.  Timbre is not just a spectral envelope, but also includes e.g. temporal dynamics like ADSR.

Spelling/grammar:
S1P3L4 laborius -> laborious
S1P3L5 bypassses -> bypasses
S3.2P2L-1 due -> due to
S4.4P2L1 twice as better than -> twice as good as
S4.4P2L3 basline -> baseline

---

> ### Author Response · Authors · 2018-11-21
> **In relation to minor comments**
>
> "Minor points / nitpicks:
>
> One of the claimed contributions is dithering before quantization to 8-bits.  How does this compare to using mixtures of logistics as in Salimans et al. 2017?"
>
>  Thank you for your comment. We were limited by the page limit and left this for future work.
>
>
> "S2P3 claims SynthNet does not use note velocity information; this is stated as an advantage but seems to make the task easier while reducing applicability to "real" music."
>
> Thank you for your comment. SynthNet produces similar results to Engel et al (2017) despite not using any velocity information. This does not make the task easier since the labels are less informative.
> The only pitch information is the note on / off times.
>
>
> "S4P1 and S4.1P4 state MIDI is upsampled using linear interpolation.  What exactly does this mean?  Also, the representation is pianoroll if I understand correctly, so what does it mean to say that each frame is a 128-valued vector with "note on-off times"?  My guess is it's a standard pianoroll with 0s for inactive notes and 1s for active notes, where onsets and offsets contain linear fades, but this could be explained more clearly."
>
> Thank you for your comment. Your understanding is correct. No linear interpolation is needed, rather we forward fill the binary midi data to obtain our temporal grid version at the appropriate sampling rate. During the piano roll data construction, we simply treat the midi note on/offs as points in continuous time and look up whether the note is on at the given grid point, where the grid may be arbitrarily fine grained. One could call this “forward filling” rather than “linear interpolation”.
>
> "What is the explanation of the delay in the DeepVoice samples?  If correcting this is just a matter of shifting the conditioning signal, it seems like an unfair comparison."
>
> Thank you for your comment. This is not just a matter of shifting the conditional signal. It is possibly due to initialization (zeros, the same across all models) and more importantly due to the autoregressive process at generation time which causes the generated series to diverge abruptly once a major mistake is made. This results in other types of errors / audio artefacts, not only delay but also playing the wrong pitch, sustained pitches or just skipping pitches altogether. Furthermore, the generation process is inherently stochastic so even though the generated signals are very close to the ground truth, they will be different every time even though the same tune is played (the differences can be observed at a ‘microscopic’ level) but the big picture, overall shape of the waveforms are visibly very close.
> SynthNet learns a mapping from pitch to fundamental frequency in the first layer, unlike the baselines where all dilated layers are conditioned as in Van Den Oord et al. (2016). We also use an auxiliary learning task on the midi component. These two changes result in fewer errors at generation time.
>
>
> "S1P2 points (2) and (3) arguing why music is more challenging than speech are questionable.  The timbre of a real musical instrument may be more complex than speech, but is this true for SoundFonts where the same samples are used for multiple notes?  It's not clear what the word "semantically" even means with regard to music."
>
> Thank you for your comment. Yes, it is true, and this is also because multiple notes can overlap. Semantically can be understood as the melodic component or the content / song if a parallel is made between music and language.

---

> ### Author Response · Authors · 2018-11-21
> **In relation to your fourth comment**
>
> "Another improvement would be to perform an ablation study over the many architecture changes.  This idea is mentioned in 4.2 but seemingly dismissed due to the impracticality of performing listening studies, which motivates the use of RMSE-CQT.  However, no ablation study is actually performed, so it's not obvious what readers of the paper should learn from the new architecture even restricted to the domain of SoundFont-rendered music generation."
>
> We did compare different hyperparameters and selected the best for all methods (Table 2). Due to space constraints, we could not include further experiments. However, the comparison between the baselines and SynthNet can be considered as an ablation study since the baselines have a different architecture (conditioning on every layer, no separable depthwise convolutions, no auxiliary task, etc). We did not include separate experiments for each change in the architecture due to space constraints.
> SynthNet learns a mapping from pitch to fundamental frequency in the first layer, unlike the baselines where all dilated layers are conditioned as in Van Den Oord et al. (2016).
> The effect of the other architectural changes can be inferred from the referred literature.

---

> ### Author Response · Authors · 2018-11-21
> **In relation to your third comment**
>
> "One change that could potentially increase the paper's impact is to train and evaluate the model on MusicNet (https://homes.cs.washington.edu/~thickstn/musicnet.html), which contains 10+ minutes of recorded audio and aligned note labels for each of ~5 single instruments (as well as many ensembles).  This would provide evidence that the proposed architecture changes improve performance on a more realistic class of polyphonic music."
>
> Thank you for your comment. We were restricted by the page limit. From our initial exploration of the MusicNet dataset, we observed that the recording conditions differ and the quality varies. The alignment is also not fully consistent. While varying conditions would be desirable for a classification task like timbre recognition, this would not be beneficial for learning synthesisers, where consistency is required. Furthermore, since the recording conditions and quality varies in MusicNet, this implies that more data would be needed to account for the differences (e.g. same note played by different violins in different acoustical conditions would result in entirely different waveforms).

---

> ### Author Response · Authors · 2018-11-21
> **In relation to your second comment**
>
> "It isn't clearly stated until Section 4 that the goal of the work is to model SoundFont-rendered music.  (The title "SynthNet" is suggestive but any music generated by such an audio model could be considered "synthesized".)  Using a SoundFont instead of "real" musical recordings greatly diminishes the usefulness of this work; adding and concatenating outputs from the single-note model of Engel et al. removes any real need to model polyphony, and there's no compelling argument that the proposed architecture changes should help in other domains. "
>
> We achieve similar results to Engel et al (2017) (their data is also synthetic) with less data, faster training and with the potential for further expressiveness. We believe that it is important to validate the work on simpler datasets, in controlled conditions, before proceeding to further expressiveness (i.e. using real annotated music).
> In their work, Van Den Oord et al. (2016) revolutionized text to speech. However, while the speech waveforms sound identical, they are very different to the actual ground truth speech. SynthNet produces waveforms that are very similar to the ground truth (Fig. 4 and Appendix 1) as demonstrated for 7 virtual instruments. Our method overcomes both challenges of capturing the instrument timbre while accurately reproducing the content – the latter being more challenging. The near-exact generated waveforms implies that high fidelity series can be very accurately generated. This enables fine control over the generative process in any domain.

---

> ### Author Response · Authors · 2018-11-21
> **In relation to your first comment**
>
> "The experimental results and provided samples do clearly show that the proposed architecture does well at reproducing the sounds of the training instruments for new MIDI scores, as measured by CQT error and human preference.  However, the fact that the model is able to nearly-exactly reproduce CQT is contrary to intuition; given only note on/off times, for most instruments there would be many perceptually-distinct performances of those notes.  This suggests that the task is too heavily restricted."
>
> Thank you for your comment. Indeed, there would be many perceptually-distinct performances of those notes if the score / piano roll would be distinct. The CQTs compared are based on waveforms of the synthesiser and of SynthNet and baselines, therefore they should be nearly-exact. Any variation in the ground truth should be reproduced by a synthesiser-learning architecture. We were the first to show that it is possible to produce nearly-exact waveforms. We show this using direct measurements, MOS tests and also provide links to all the samples used in all experiments.
> ( http://bit.ly/synthnet_mostest  http://bit.ly/synthnet_table3  http://bit.ly/synthnet_appendix_a )
> We consider this to be a fundamental and necessary step towards further expressiveness of learned synthesisers.

---

### Meta-Review · Area_Chair1 · 2018-12-13
**evaluation on synthetic data is a major limitation**

**Confidence:** 5
**Recommendation:** Reject

**Metareview:**

The paper describes a WaveNet-like model for MIDI-conditional music audio generation. As noted by all reviewers, the major limitation of the paper is that the method is evaluated on a synthetic dataset. The rebuttal and post-rebuttal discussion didn't change the reviewers' opinion.